# Equine Distal Limb Wounds: Economic Impact and Short-Term Prognosis of Non-Synovial Versus Synovial Lesions in Southern Germany

**DOI:** 10.3390/vetsci12030205

**Published:** 2025-03-01

**Authors:** Valeria Albanese, Paola Straticò, Holger Fischer, Lucio Petrizzi

**Affiliations:** 1Tierärztliches Kompetenzzentrum für Pferde Großwallstadt Altano GmbH, Niedernberger Str. 9, 63868 Großwallstadt, Germany; vzalbanese@gmail.com (V.A.); h.fischer@pferdeklinik-grosswallstadt.de (H.F.); 2Department of Veterinary Medicine, Località Piano D’Accio, 64100 Teramo, Italy; lpetrizzi@unite.it

**Keywords:** wound, distal limb, synovial cavity, synovial infection, horses

## Abstract

Injuries to the limbs are a common occurrence in horses. Sometimes joints, bursas, or tendon sheaths, so called synovial structures, may be involved in limb wounds, too. This carries the risk of infection of such structures, which may have a clinical and financial impact. Additionally, tendons and/or ligaments may be involved. The aim of this study is to compare wounds with and without involvement of synovial structures, and wounds with and without tendon and/or ligament involvement, as far as cost of treatment and clinical outcomes.

## 1. Introduction

Injuries to the distal limb are common in horses. The clinical aspect of the lesions is variable based on which structures are involved. The anatomy of the musculotendinous apparatus of equine distal limb is made complex by the presence of synovial and non-synovial structures very close to each other and poorly protected by surrounding tissues [1]. The involvement of either of the two, or both, affects the prognosis as well as the type of intervention. The presence and proliferation of bacteria within a synovial cavity leads to severe inflammation and secretion of inflammatory mediators that directly cause modification of the internal homeostasis of the injured synovial cavity, and metabolic changes with secondary damage to the articular surface or teno-ligamentous structures, with secondary osteoarthritis and adhesions [1,2]. Early recognition of involved structures is mandatory to establish an appropriate treatment and improve prognosis [3,4,5].

A multimodal approach is suggested when a synovial lesion is suspected. Treatment aims to remove pathogen and inflammatory mediators [6]. Also, non-synovial wounds need to have a prompt intervention, to avoid migration of the microorganism into the adjacent synovial cavity [7]. Systemic broad-spectrum antibiotics are usually combined with local intravenous limb perfusion (IVLP). Surgical debridement and synovial lavage are recommended to remove debris and devitalized tissue, and enhance vascularization and local antibiotic diffusion [8,9].

Prognosis is usually good for survival: according to published literature, 85% of horses with synovial infections survive to discharge. The percentage of success decreases when return to athletic function is evaluated [10,11,12,13].

How quickly treatment is initiated is deemed important, with best prognoses associated with a time frame of 24–36 h from the initial trauma [11,14,15]. Economic constraints may play a role in the decision process on what treatment is affordable, with direct effects on the prognosis.

This retrospective study was conducted to investigate clinical outcomes of horses referred for limb wounds for which surgical treatment was elected.

The primary objective was to assess and compare the clinical and economic impact of synovial versus non-synovial limb wounds in horses. Key metrics included duration of hospitalization, total and daily hospitalization costs, number of surgeries, duration of antibiotic treatment, and survival to discharge. Secondary objectives were to differentiate outcomes between complicated and non-complicated cases within the non-synovial group, to compare the economic impact of complicated non-synovial wounds to those involving one or more synovial structures, and to examine the association of the use of IVLP with survival in synovial cases. (Table 1).

## 2. Materials and Methods

### 2.1. Case Definition and Selection

Study population consisted of horses referred to a private equine hospital located in southern Germany for limb wounds between 2021 and 2023. Inclusion criteria were: age older than 1 year, one or multiple limb lacerations, surgical treatment under general anesthesia. Additional injuries such as fractures, or tendon or ligament transections of subjectively more than 25% of the anatomical structure’s cross-section, were considered exclusion criteria.

Horses were classified into two cohorts, as determined by clinical evaluations and diagnostics. Cohort 1 (synovial) included horses with wounds involving synovial structures. All horses in Cohort 1 underwent one or more arthroscopic lavage. Cohort 2 (non-synovial) included horses with wounds not involving synovial structures. Cohort 2 was further divided into complicated and non-complicated cases. Cases were defined as complicated when one or multiple tendons and/or ligaments were involved in the laceration.

### 2.2. Data Collection

Data were retrospectively collected from medical records. Signalment, time to referral, and location of the wound were recorded. The evaluated variables included duration of hospitalization, total and daily cost of hospitalization, number of surgical procedures performed, duration of antibiotic treatment, use of intravenous limb perfusion (IVLP), and survival to discharge. The use of intravenous regional limb perfusion in Cohort 1 and its association with survival outcomes were also analyzed. Additionally, time elapsed between wound detection and referral (time to referral) was examined across both cohorts to assess its association with survival, cost, duration of hospitalization, and number of surgical procedures.

### 2.3. Statistical Analysis

Comparisons were made between synovial and non-synovial groups and between non-synovial groups for complicated versus non-complicated cases. Fisher’s exact test and a Pearson’s Chi-squared test were used to investigate categoric variables. A Wilcoxon rank sum test was used to compare groups for continuous variables. The association between survival to discharge and intravenous regional limb perfusion was evaluated using the Chi-square test. A linear and logistic regression assessed the impact of the cost per day, hospitalization length, and the number of surgeries on time to referral for both cohorts, respectively, for continuous and categorical variables. Statistical significance was set for *p* < 0.05.

## 3. Results

A total of 51 horses were included in the study. For 21/51 (41%) of them, involvement of at least one synovial structure was confirmed. These horses were allocated to Cohort 1. Thirty (59%) did not show any synovial involvement; therefore, these horses were allocated to Cohort 2.

In the synovial cohort, 24 structures were involved in 21 horses (Table 2). In three horses, multiple structures were involved.

All 30 horses in the non-synovial cohort survived to discharge (100%). Of the 21 horses in the synovial cohort, 18 survived to discharge (85.7%).

In Table 3, the differences between the two Cohorts in the number of surgeries, duration of antibiotic treatment, total hospitalization costs, length of hospitalization, cost per day, and survival to discharge are summarized.

There were notable differences in hospitalization metrics within Cohort 2, when comparing complicated and non-complicated cases (Table 4): a significantly longer median length of hospitalization for complicated cases, and a higher hospitalization cost.

Time to referral ranged between 6 and 42 days, with a median value of 7 days (with a median time for both cohorts). Univariate regression analysis was employed to investigate how different outcomes were associated with time elapsed between diagnosis and referral (time to referral) in both cohorts (Table 5). In Cohort 1, there was a significant association found between time to referral and cost per day (coefficient: 0.94, *p* < 0.001). This is a relationship that was not observed in Cohort 2. No significant association between time to referral and hospitalization length, or between time to referral and number of surgeries was noted in either group (Table 6).

The study also examined the association of survival to discharge with the use of IVLP in synovial cases. A Chi-square test found no significant association between these variables (*p* = 0.181) (Table 6).

A reliable analysis between ‘survival to discharge’ and ‘duration prior to referral’ was not possible due to a significant imbalance of data. For the same reason, it was not possible to evaluate possible differences in prognoses within Cohort 1 related to which structure was involved.

## 4. Discussion

### 4.1. Multiple Surgical Procedures

Contamination of one or multiple synovial structures may warrant multiple surgical procedures to optimize the chances of a full recovery. According to the previous literature [16], contaminated or infected synovial structures need to be lavaged promptly, aggressively, thoroughly and, at times, repeatedly. In this dataset, a higher percentage of horses with synovial involvement (28.6%) needed multiple procedures than horses without such involvement (6.7%). Multiple procedures consisting in joint lavages were deemed necessary to address synovial contamination in those cases. Although a multimodal approach to the treatment of wounds other than antimicrobials in horses is described [17], in this cohort study the antibiotic therapy still showed a high impact on the medical strategy, with longer treatment in the synovial cohort.

A thorough report and analysis of systemic antimicrobial treatment protocols go beyond the purpose of this work. However, it is worth mentioning that most horses in this report received an initial course of 5–7 days broad-spectrum antibiotic combination (Procain penicillin 22,000 IU/kg IM BID or Amoxicillin 10 mg/kg IV BID, and Gentamicin 6.6 mg/kg IV SID) followed by a course of oral antibiotic (Trimethoprim-Sulfadiazin 30 mg/kg PO BID or Doxycyclin 10 mg/kg PO BID or Enrofloxacin 7.5 mg/kg PO SID), depending on the results of culture sensitivity tests. Treatment was administered if it was deemed clinically necessary for each individual case.

Most horses with synovial involvement (71.4%) only needed one lavage to resolve. Synovial lavage was performed via arthroscopy, tenoscopy, or bursoscopy. Compared to through-and-through needle lavage, the endoscopic technique offers many advantages. The operator may benefit from visualization of the affected structure and foreign material, and fibrin pannus or debris may be removed with instruments. In addition, higher fluid volume and pressure can be utilized. Disadvantages of arthroscopic technique are the need of specialized instrumentation and skills, as well as the costs of the procedure [2,12,18,19]. The choice of arthroscopic lavage over needle through and through lavage may have also contributed to treatment success after only one procedure.

Furthermore, referral and subsequent treatment were mostly carried out promptly after diagnosis. Therefore, most affected synovial structures were likely lavaged before contamination turned into an established infection, which may have also contributed to the success rate of only one lavage procedure.

### 4.2. Time to Referral

The literature suggests that timely intervention leads to a better prognosis in cases of synovial contamination; the past literature has mostly supported this claim [8,15,20,21]. Contamination of a synovial structure does not equal infection of said synovial structure based on the presence or proliferation of bacteria [22]. However, untreated contamination may turn into an infection and affect the prognosis [10,11,12,23,24].

No additional diagnostics were consistently carried out to differentiate between contaminated and infected structures in our data.

The distribution of data made a statistical evaluation of the association between time to referral and survival not possible. Nevertheless, in both cohorts the survival rate was close to 100% (85.7% in Cohort 1; 100/in Cohort 2) despite the wide range of time to refer.

A significant positive association was observed between the time to referral and daily costs of treatment (*p* < 0.001) for horses in the synovial cohort. This association was not significant for non-synovial cases, where the impact of time to referral on daily costs was not significant (*p* = 0.4).

There may be several reasons for this. The initial assessment of a wound with suspected synovial involvement may be more expensive [25,26] in terms of time and diagnostics. The use of larger volumes of fluids while lavaging a synovial structure with an infection of several days’ duration may also result in higher costs. Unfortunately, a breakdown of the bill was not performed to further investigate these possibilities. In any case, this finding indicates that a longer waiting time before referral could be economically detrimental. Many times, the very reason for delaying referral is, indeed, the perception that there would be a more thrifty and just as successful treatment modality in the field. However, if referral is a necessity, delaying it may end up in even higher veterinary costs.

The lack of correlation between time to referral and length of hospitalization (*p* = 0.5), as well as the lack of correlation between time to referral and the number of surgical procedures performed (*p* = 0.5 for synovial cohort, *p* = 0.8 for non-synovial cohort) align with the previous literature claiming that delaying time to referral may not have a clinically negative impact on either of them [13,19,22,27,28]. The lack of impact of time to referral on these parameters reflects that the chosen treatment defeated the infection just as effectively, irrespectively of the duration of the infection itself. This could be due to the type of bacteria involved and their antibiotic susceptibility. All lacerations occurred at home, and not in a hospital setting. Therefore, infections were most likely caused by wild-type bacteria, for which antibiotic resistance is not as common as it is for nosocomial bacteria [29]. Another possible reason is the aggressive initial surgical debridement; all horses involved in this study underwent major surgical wound debridement with or without synovial lavage. The debridement could have been radical enough to excise most infected tissue even in long standing cases.

### 4.3. Length of Hospitalization

This population of horses consists predominantly of low- and medium-level sport horses, owned by non-professional equestrians. Many horses in the area are kept in paddocks or pastures and cannot be confined to a stall or box at home. Although no data are available, it is the authors’ opinion that the combination of the lack of sufficient clients’ knowledge and expertise and the lack of facilities to confine convalescent horses in at home may lead to longer hospitalizations.

Horses in both cohorts were hospitalized for a similar length of time (range 7–13 days, median 10 days for the non-synovial cohort; range 8–17 days, median 11 days for the synovial cohort). One of the factors that leads to this finding may be that most horses in the synovial cohort only required one surgery. On the other hand, the lack of significant difference could also be because horses in the non-synovial cohort were hospitalized longer than strictly necessary to accommodate owner’s needs and preferences.

### 4.4. Treatment Cost

The total cost of treatment was significantly higher for horses in the synovial cohort (range EUR 5050–9662, median EUR 5979) than for horses in the non-synovial cohort (range EUR 2746–6489, median EUR 4384) (*p* = 0.003). Daily costs of treatment were significantly higher for the synovial than for the non-synovial cohort. Factors involved in this difference were higher surgical fees for the arthroscopic lavage compared to wound debridement and care, as well as material used and level of post-operative care and diagnostics, such as repeated synoviocentesis to help assess recovery.

### 4.5. Prognosis

All horses in the non-synovial cohort and most horses in the synovial cohort survived to discharge. Three horses in the synovial group were euthanized before discharge. One had a wound that caused an infection of the elbow joint; the patient was referred 42 days after the wound occurred, and, despite aggressive repeated lavage, remained lame, and was, therefore, euthanized. The second non-surviving horse had both distal intertarsal and tarsometatarsal joints involved. Despite IVLP and needle through-and-through lavage of the joints performed twice under general anesthesia, the horse’s level of comfort deteriorated, and was therefore euthanized. Lower hock joints are not amenable to arthroscopic lavage due to their anatomy. Therefore, a through-and-through needle lavage was performed. This technique allows only limited amounts of fluids through the joint space and, since an instrument portal is not present, does not allow for debridement of any necrotic material or pannus. Therefore, it is less effective in removing all the infection from the joint space, which might have been the reason for failure in this case. The third non-survivor was a yearling that, after successful treatment of a tarso-crural sepsis following a laceration, developed a septic physitis in another limb and was euthanized. The pathogenesis of septic physitis was most likely hematogenous spread from the septic joint. Treatment could have been implemented, consisting in intravenous regional limb perfusion and direct antibiotic treatment of the physis self; however, any further treatment was declined.

All three non-survivors had concurrent factors that made their clinical course inherently more complicated than a simple laceration with synovial involvement: in one case the wound was over 6 weeks old. In the second case, the joint was poorly accessible for treatment. In the third case, an additional, most likely hematogenous infection presented itself at a different, remote site.

Treatment of septic synovitis in horses has very broad survival rates ranging from between 56% and 100% [9,10,13,15,19,20,21,28,30,31,32,33,34,35,36]. The prognosis for short-term survival in the synovial cohort (89%) is at the high end of that range. Nevertheless, the prognosis for survival of the synovial cohort was lower than that of the non-synovial cohort (100%). The difference approximated but did not quite reach statistical significance (*p* = 0.064). These data are in accordance with previous studies [37], despite the potential selection bias that this study could have suffered. In fact, only horses undergoing surgery were enrolled, leading to the exclusion of those horses that have been euthanized before referral for economic of perceived poor prognosis. This may have overestimated the survival rate and therefore the prognosis.

### 4.6. Comparison Complicated Non Synovial/Synovial

The median cost of treatment for cases without synovial involvement but with tendons and/or ligaments lesions was significantly higher than for cases without such involvement, indicating a substantial financial impact of such finding.

The number of surgical procedures under general anesthesia and the total length of hospitalization were not significantly different among the two subsets.

This finding is especially relevant at the time of formulating an estimate in terms of time and costs before undertaking treatment. It appears, then, that the involvement of tendons and ligaments contributes to much higher costs and longer treatment.

Unfortunately, due to the retrospective nature of the study, data about the return to athletic function was missing in too many cases to render statistical analysis of this variable not possible.

Additional limitations of this study are its small sample size, grouping of all synovial structures in one cohort, lack of some data such as culture and sensitivity results and return to function, and its possible aforementioned selection bias.

## 5. Conclusions

Traumatic wounds of the equine distal limb in horses carry a good prognosis for survival in the geographic area where the study was performed. This may be partly due to management practices and referral practices in southern Germany and may not necessarily reflect the reality of other geographical areas. In the case of synovial involvement and non-synovial lesions involving a tendon or a ligament, owners in this area must be informed about the high treatment costs. In the case of synovial lesions, a multimodal treatment based on synovial lavage under general anesthesia with arthroscopic guidance, and systemic and local antibiotics must be set up as soon as possible. When the lesion is not penetrating a synovial structure, surgical debridement and antimicrobial therapy should be advised. Although no association between time to referral and prognosis was found in this research, prompt intervention and referral are advisable to reduce the chances of an infection developing.

## Figures and Tables

**Table 1 vetsci-12-00205-t001:** Objectives of the study.

1. Objective	2. Objective	3. Objective	4. Objective
Synovial vs. non-synovial limb wounds	Complicated vs. non-complicated non-synovial wounds	Complicated non-synovial wounds vs. synovial wounds	Association of IVLP with survival in synovial cases

**Table 2 vetsci-12-00205-t002:** Synovial structures involved in Cohort 1.

Structure	Number
Digital flexor tendon sheath, hind	7
Tibio-tarsal joint	4
Metatarsophalangeal joint	2
Metacarpophalangeal joint	2
Radial carpal joint	2
Digital flexor tendon sheath, fore	1
Navicular bursa, fore	1
Elbow	1
Calcaneal bursa	1
Distal intertarsal joint	1
Tarsometatarsal joint	1
Tarsal sheath	1
**Total**	**24**

**Table 3 vetsci-12-00205-t003:** Comparative analysis between synovial vs. non-synovial joint involvement cases.

	Non-Synovial, *n* = 30 ^1^	Synovial, *n* = 21 ^1^	*p*-Value ^2^
Number of Surgeries			
1	28 (93.3%)	15 (71.4%)	0.040
2	2 (6.7%)	3 (14.3%)	
3	0 (0%)	3 (14.3%)	
Duration of Antibiotic Treatment (days)	16 (12–19)	20 (17–25)	0.018
Total Hospitalization Cost (euro)	4.384 (2.746–6.489)	5.979 (5.050–9.662)	0.003
Length of Hospitalization (days)	10 (7, 13)	11 (8, 17)	0.3
Cost Per Day (euro)	403 (330–499)	536 (424–652)	0.030
Survival to Discharge (number of horses)	30 (100%)	18 (85.7%)	0.064

^1^ *n* (%); Median (IQR), ^2^ Fisher’s exact test; Pearson’s Chi-squared test; Wilcoxon rank sum test; Wilcoxon rank sum exact test.

**Table 4 vetsci-12-00205-t004:** Non-synovial group complicated versus not complicated length of hospitalization, hospitalization total and daily cost analysis.

	Complicated, *n* = 5 ^1^	Not Complicated, *n* = 25 ^1^	*p*-Value ^2^
**Length of Hospitalization (days)**	24 (13, 26)	10 (7, 12)	**0.024**
**Total Hospitalization Cost (Euro)**	8000 (6494–8199)	3834 (2675–5041)	**0.016**
**Cost Per Day (Euro)**	342 (308–475)	405 (346–500)	0.6

^1^ Median (IQR); ^2^ Wilcoxon rank sum test; Wilcoxon rank sum exact test.

**Table 5 vetsci-12-00205-t005:** Univariate regression analysis for association of different factors with the time to referral for synovial and non-synovial cases.

Variable	Cohort 1			Cohort 2		
	Coefficient ^1^	95% CI	*p*-Value	Beta ^1^	95% CI	*p*-Value
Cost per Day	0.94	0.52, 1.4	<0.001	1.0	−1.5, 3.5	0.4
Total Hospitalization Cost	−0.49	−7.4, 6.5	0.9	−0.52	−27, 26	>0.9
Hospitalization Length	−0.03	−0.09, 0.04	0.5	−0.03	−0.13, 0.07	0.5
Number of Surgeries	0.00	−0.01, 0.00	0.5	0.01	−0.09, 0.05	0.8

CI = Confidence Interval. Note: Linear regression was used for the cost per day, total hospitalization cost, and hospitalization length (coefficient = Beta) while logistic regression was used for number of surgeries (coefficient = Odds ratio). ^1^ Coefficient is the Odd Ratio resulting from the the linear or logistic regression. Beta is the same.

**Table 6 vetsci-12-00205-t006:** Chi-Square Test for association of survival-to-discharge with intravenous regional limb perfusion ^1^.

Chi-Square Test for Association Survival to Discharge and IVRLP		
	IVRLP Given ^2^	IVRLP Not Given ^2^
Survivors (number of horses)	0	3
Non-survivors (number of horses)	11	7

^1^ Chi-square statistic = 1.79, df = 1, *p*-value = 0.181. ^2^ Pearson’s Chi-squared test with Yates’ continuity correction.

## Data Availability

Data supporting the findings of this study are stored securely at the institution. Due to confidentiality agreements and ethical restrictions, the raw data are not publicly available but can be provided upon reasonable request to the corresponding author.

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
