# Peer review of "Equine Distal Limb Wounds: Economic Impact and Short-Term Prognosis of Non-Synovial Versus Synovial Lesions in Southern Germany"

_vetsci, 2025, doi:10.3390/vetsci12030205_

Round 1
Reviewer 1 Report
Comments and Suggestions for Authors
Overall, this manuscript presents interesting findings on lower limb wounds in horses. Although, some of the findings are generally regarded as common knowledge, it is nice to see some metrics and statistical analysis on a decent number of cases. The article is generally well written, but there are a few sentences that have an awkward construction and will need to be rephrased. There is some information that is mentioned only in the discussion, that should have been presented in the results, so please address this issue as per comments below. Also, the authors have discussed that their hypotheses were either confirmed or rejected in the discussion. Unfortunately, there is no mention of any hypotheses earlier in the manuscript, as it should be. Generally, hypotheses are presented in the last paragraph of the introduction, together with the objectives. You will need to either state your hypotheses there, or alternatively just don’t have any hypotheses and discuss the findings in the discussion without mentioning hypotheses. It is not mandatory to have hypotheses, it doesn’t really add any value to the quality of manuscripts in general. Please consider the suggestions below:
Line 40:” not-synovial” is incorrect, either use “non-synovial”, or even better just describe these wounds as “wounds without synovial involvement”.
Lines 52-53: “the economic issue is often a reason for the owner to reject treatment recommendation, with inevitable effects on the clinical resolution of the condition.” This doesn’t sound very good, please rephrase, particularly “the economic issue” and “inevitable”, perhaps it is a literal translation from another language?
Line 55: “major surgical treatment”, lavage is not that major, just use “surgical treatment”
Line 56: “The primary objective…”, also use ”impact”, not plural.
Lines 60-61: this sounds confusing, please rephrase.
Line 65: “Population and Sample Selection”, horses and wounds are not exactly samples, perhaps just use “Case definition and selection”
Line 68: major surgical; implemented use performed instead.
Line 70: more than 25% of the anatomical structure’s cross-section
Line 76: were those WHERE…..
Lines 77-78: it’s a repeat of lines 69-71. You have not specifically defined non-complicated cases. Even though it is somewhat implicit that they are the opposite of complicated cases, you should still define what non-complicated cases were in this study.
Line 89: “referral date and time (hours)”, do you mean time to referral?
Lines 80-92: there are repetitions in the “Data collection” and “Data variables”, two separate paragraphs are not justified as you end up repeating things. Combine these 2 paragraphs.
Line 94: change to “Comparisons were made”
Line 96: categorical, not categoric
Line 100: “pre-arrival duration”? Do you mean again time to referral? Or call it “time from injury to referral” or “time from injury to admission”
Line 101: I think you need to define what the statistical significance is for you in your study. I assume it’s p<0.05, but needs to be defined in the methods.
Table 1: since you differentiate between hind and forelimb digital sheath, I feel you also need to specify if the navicular bursa was hind or forelimb.
Lines 109-110: the second line should be adjacent to the first line, not below.
Lines 112-113: the second line should be adjacent to the first line, not below. I think that the sentence at lines 111-112 can actually be deleted as you discuss that just below.
Lines 125-126: change to “though the difference was not statistically significant”
Table 2: the percentages in the number of surgeries don’t add up. If you decide to use decimals (i.e. 6.7%) then you have to be consistent and use 93.3% (line 114 will need to be corrected accordingly as currently it says 93%). Same for the percentages in the synovial column (currently if you add them up you get 99%. Another thing in Table 2 that may need correction (if I interpreted things right) is where the p value for the number of surgeries has been placed (0.040). Shouldn’t 0.040 be in the row of 1 surgery rather than the whole category “number of surgeries”?
Table 5 lines 162-163: I am not sure I understand “Survival-to-Discharge Duration…..”. What duration are you referring to? What does duration have to do with that? Also the table seems to be missing information on what the rows are. I am assuming the top row is the number of non-survivors and the bottom row are the survivors, but you need to add that.
Line 170: procedures, plural. Also, delete “It is common knowledge” and just rely on references please.
Line 172: “as many times as needed” probably not the best language, you may want to change to “and, at times, repeatedly”.
Line 176: What hypothesis? The is the first time in the entire manuscript where the word hypothesis has been used. It’s better to just describe your findings than trying to build hypothesis after the fact.
Line 177: 72% doesn’t match table 2. In the table you wrote 71% (which you should actually change to 71.5% (15/21) to be precise. Line 114 will need to be corrected accordingly.
Line 185: you are missing a full stop after “costs of the procedure [2,9,14,15]”
Line 186: add needle to “through and through lavage”
Line 188: carried out
Line 191: success rate with only. Also, you now use lavage, previously you used flushes. Probably better to use lavage rather than flush, flush is more informal
Line 192: time to referral
Lines 193-194: I think this could just be rephrased using just the literature, please delete “experience and common sense”. There is no space for common sense in scientific literature.
Lines 199-200: this is new information that I couldn’t find in the results section. The results is where you should place data, the discussion part is for discussing not for presenting new data. Also is it correct to say “diagnosis” at line 199? If there was a diagnosis of synovial sepsis why would a vet wait 42 days to refer it? Did you mean “time of injury” rather than diagnosis? It is also not clear if you are referring to wounds with synovial involvement or non-synovial. These 2 lines need some work.
Lines 202-203: time to referral
Line 203: instead of meaningless, perhaps use “not possible”.
Lines 204: time to referral
Line 205: again there was never a hypothesis anywhere in the manuscript, so I am not sure why you now have come up with a second hypothesis in the discussion.
Lines 206: time to referral
Line 207: “This association isn’t significant”….avoid contracted verb forms in scientific literature. Also be consistent with verb tenses, in the previous sentence you used the past tense, change to ” This association was not significant…”
Lines 208: please delete “where the impact of time to refer on daily costs isn’t clear”, it is significant or it is not, in a way it is very clear.
Line 219: time to referral
Line 220: time to referral
Line 222: “may not have a clinically negative impact”. On what? You need to define that, is it on length of hospitalization and number of surgeries? If so, then just stop the sentence after “align with previous literature.”
Line 228: “nosocomial bacteria.” Add a reference.
Lines 233-234: “Before discussing this point, a few forewords on the client and horse population of the institution where data have been collected are necessary.” Not necessary.
Line 238-239: “leads to longer hospitalizations compared to other geographical areas.” Do you have data to show that? If not, delete the sentence. Certainly there wasn’t any data on that in your manuscript.
Line 240: again what hypothesis? Your manuscript has no hypotheses.
Lines 248-249: be consistent, either use the symbol of euro as you have done previously in the manuscript, or spell it out everywhere.
Line 249: same again for the hypothesis. I have never a paper with so many hypotheses to start with, let alone the fact that they haven’t been stated anywhere other than in the discussion, which is not where that should be done.
Line 261: add needle before “lavage”
Line 279: “Evaluating septic synovitis in horses have found”, not grammatically correct, please rephrase. Also you need a reference here.
Line 285: “Infact”, In fact,…..
Line 294: you used “major surgical procedures” many times in the manuscript, but I am not sure there is a well-accepted definition of what is considered major, it’s quite subjective. I wonder if it wouldn’t be better to just use “surgical procedures under general anesthesia.”
Line 304: can’t could not
Line 306: “higher” than what?
Line 310: “we didn’t highlight”, change to “we did not find” (or identify)
Reviewer 2 Report
Comments and Suggestions for Authors
Dear authors,
It is a pleasure for me to review this manuscript, which addresses a topic of great relevance in equine medicine, given the high frequency of occurrence and the challenge of treating distal limb injuries. My comments are below:
INTRODUCTION
The introduction could be improved with the use of more current references. Both the introduction and discussion are based on old bibliography and an update could further enrich the manuscript.
RESULTS
- When analyzing the results in Table 3, the question arose whether the data from the 5 complicated cases (non-synovial group) were included or not in the comparison between synovial x non-synovial groups (Table 2). Please clarify this point.
- Why were the results regarding the time to referral not presented? It would be interesting to present this information.
DISCUSSION
- It was missing and it would be interesting to discuss the results regarding the duration of antibiotic treatment.
- Lines 173, 174 and 177: the values ​​described in the discussion are not exactly the same as those presented in the results (lines 113 and 114). Please keep the same values ​​throughout the text.
- Line 175: “Our first hypothesis was, therefore, confirmed.” It is suggested to present the study hypotheses at the beginning of the manuscript.
- Line 199: “In our dataset, time elapsed between diagnosis and intervention ranged between 6 hours 199 and 42 days, with a mean of 7 days.” This data must be presented in the results session.
- Lines 203, 204: “Nevertheless, in both cohorts’ survival rate was close to 100% (89% in cohort 1; 100/ in cohort 2) despite the wide range of time to refer.” The result presented in line 110 was 85.7 and in Table 2 it was 86. Please keep the same information throughout the text.
- Line 204: “Our second hypothesis was, therefore, rejected based on the data.” This hypothesis was not presented at the beginning of the manuscript.
- Line 240: “Opposite to our hypothesis, horses in both cohorts were hospitalized for a similar length…”. This hypothesis was not presented at the beginning of the manuscript.
- Line 241: “…(range 7-13 days, mean 10 days for the synovial cohort; range 8-17 days, mean 11 days for the non-synovial cohort).” This information is the opposite of what appears in Table 2.
Are these results mean or median? The discussion refers to the mean, but in the Table 2 it appears as median.
- Line 248: “…(range 5,050-9,662 Euro, mean 5,979) than for horses in the non-synovial cohort (range 2,746-6,489, mean 4,384) (p = 0.003)…”. Again: Are these results mean or median? The discussion refers to the mean, but in the Table 2 it appears as median.
- Line 249: “…confirming our hypothesis.” Again: This hypothesis was not presented at the beginning of the manuscript.
- Line 279: “Evaluating septic synovitis in horses have found very broad survival rates ranging between 56% and 100%.” The reference for this information was missing.
Reviewer 3 Report
Comments and Suggestions for Authors
To Authors and Editors
Dear Sirs,
concerning your manuscript, I feel that, although, the text is well-written, I could not find a new piece of information that could add something to the existing literature. Moreover, as a clinician, I had quite a few queries regarding the material and methods used from a clinical point of view, meaning that I would like to see which antibiotics you used locally on systematically, which antibiotics, whether the bandaging techniques were different etc. The reason for that is that there are clinical questions arising sometimes from the lack of significant differences between your cohorts.
